# Efficient Inhibition of *Aspergillus flavus* to Reduce Aflatoxin Contamination on Peanuts over Ag-Loaded Titanium Dioxide

**DOI:** 10.3390/toxins15030216

**Published:** 2023-03-10

**Authors:** Dandan Yang, Hailian Wei, Xianglong Yang, Ling Cheng, Qi Zhang, Peiwu Li, Jin Mao

**Affiliations:** 1National Reference Laboratory for Agricultural Testing P.R. China, Key Laboratory of Detection for Mycotoxins, Laboratory of Quality & Safety Risk Assessment for Oilseed Products (Wuhan), Quality Inspection & Test Center for Oilseed Products, Key Laboratory of Biology and Genetic Improvement of Oil Crops, Ministry of Agriculture, Oil Crops Research Institute, Chinese Academy of Agricultural Sciences, Wuhan 430062, China; 2Hubei Hongshan Laboratory, Wuhan 430062, China

**Keywords:** peanuts, *Aspergillus flavus*, inhibition, visible light, Ag-loaded titanium dioxide, quality

## Abstract

Peanuts are susceptible to aflatoxins produced by *Aspergillus flavus*. Exploring green, efficient, and economical ways to inhibit *Aspergillus flavus* is conducive to controlling aflatoxin contamination from the source. In this study, Ag-loaded titanium dioxide composites showed more than 90% inhibition rate against *Aspergillus flavus* under visible light irradiation for 15 min. More importantly, this method could also reduce the contaminated level of *Aspergillus flavus* to prevent aflatoxins production in peanuts, and the concentrations of aflatoxin B_1_, B_2_, and G_2_ were decreased by 96.02 ± 0.19%, 92.50 ± 0.45%, and 89.81 ± 0.52%, respectively. It was found that there are no obvious effects on peanut quality by evaluating the changes in acid value, peroxide value, and the content of fat, protein, polyphenols, and resveratrol after inhibition treatment. The inhibition mechanism was that these reactive species (•O_2_^−^, •OH^−^, h^+^, and e^−^) generated from photoreaction destroyed cell structures, then led to the reduced viability of *Aspergillus flavus* spores. This study provides useful information for constructing a green and efficient inhibition method for *Aspergillus flavus* on peanuts to control aflatoxin contamination, which is potentially applied in the field of food and agri-food preservation.

## 1. Introduction

Industrial wastes, pesticides, heavy metals, mycotoxins, etc. are becoming more and more serious threats to food safety [1,2,3,4]. Peanuts, with rich nutrients and health value, are a type of important oil crop [5], but they are susceptible to aflatoxin contamination. Aflatoxins, with high toxicity, teratogenicity, and carcinogenicity, have been regarded as the most toxic mycotoxins found in food and agri-food [6,7,8], which not only threaten human health, but also can lead to huge economic losses [9,10,11,12,13]. Aflatoxins are secondary metabolites, mainly produced by *Aspergillus flavus (A. flavus)* in food and agri-food, especially peanuts and corn [14,15]. Therefore, controlling *A. flavus* is a meaningful approach to reducing the contamination of aflatoxins in food and agri-food from the source.

The most frequently used measures for controlling *A. flavus* include physical, chemical, and biological measures, etc. Physical measures, such as high voltage atmospheric cold plasma, and chemical measures, such as Validamycin A, could control *A. flavus* [16,17]. For biological methods, *Bacillus subtilis* QST 713 [18], *Pseudomonas stutzeri* YM6 [19], and lactic acid bacteria [20], etc. showed inhibition effects on *A. flavus*. Recently, it had been reported that photocatalysis based on semiconductors can be used as an effective way to control fungi [21].

Photocatalysis is regarded as a green, mild, and low-energy method [22]. However, exploring an efficient, low-cost, and stable catalyst that can be used in food and agri-food is challenging and significant. Titanium dioxide (TiO_2_) is a low-cost, low-toxicity, and high-stability photocatalyst, which can be also used as a food additive authorized by the Food and Drug Administration [23]. It could be used as a food additive in bakeries, chewing gums, candies, chocolates, etc. [24]. In addition, Hwang et al. evaluated the physicochemical properties, intestinal transport, and cytotoxicity of food additive TiO_2_, which were conducive to the application of TiO_2_ in the food industry [25]. It was found that TiO_2_ could inactivate *Escherichia coli* (*E. coli*) [26], *Staphylococcus aureus* (*S. aureus*) [27], and *Bacillus subtilis* [28] under ultraviolet (UV) light irradiation. However, only 5% of sunlight is UV light, which limits the practical application of TiO_2_ [29]. Ag nanoparticles (Ag NP_S_) can present the surface plasmon resonance (SPR) effect to enhance the visible light absorption region of TiO_2_ [30]. Besides, the intrinsic disinfection property of Ag NP_S_ had been proved in the field of food disinfection. Bandpey et al. made Ag-coated low-density polyethylene films, which presented promising disinfectant activity against microbes and prolonged the shelf life of milk [31]. Wu et al. reported that Ag/TiO_2_ composites showed excellent photocatalytic activity in the disinfection of *E. coli* and *S. aureus* under visible light irradiation [32]. Based on the above introduction, this study intends to combine Ag and TiO_2_ to utilize their synergistic effects for controlling *A. flavus* in food and agri-food.

Herein, Ag-loaded titanium dioxide (Ag/TiO_2_) composites were synthesized by a facile photodeposition method, and the composition and microstructure of Ag/TiO_2_ were analyzed. Then, the inhibition performance of Ag/TiO_2_ on *A. flavus* was evaluated by a modified plate colony counting method. The inhibition activity of Ag/TiO_2_ on contaminated peanuts was also estimated, and the reduction of aflatoxins concentrations was evaluated by high-performance liquid chromatography (HPLC). In addition, the effects of photocatalytic inhibition on the quality of peanuts were discussed. Finally, the reasons for photocatalytic activity enhancement were investigated by UV-visible diffuse reflectance spectrum (UV-vis DRS) analysis and photoelectrochemical tests. The photocatalytic inhibition mechanism of *A. flavus* over Ag/TiO_2_ was explored by radical trapping tests, electron spin resonance (ESR) analysis, and fluorescent spore staining tests.

## 2. Results

### 2.1. Characterization of Photocatalysts

XRD was used to study the crystal structure of as-prepared catalysts. As demonstrated in Figure 1a, the characteristic diffraction peaks of anatase and rutile were observed in TiO_2_, consistent with its mixed-crystal phases [33]. The peaks located at 25.28°, 37.80°, 48.05°, and 62.69° could be identified as the (101), (004), (200), and (204) crystal planes of anatase (JCPDS No. 21-1272), while peaks at 27.44°, 36.09°, 41.23°, 54.32°, 56.64°, and 69.01°corresponded to the (110), (101), (111), (211), (220), and (301) planes of rutile (JCPDS No. 21-1276), respectively. As for Ag/TiO_2_, an additional peak at 38.10° appeared in the enlarged XRD pattern (Figure 1b), which was ascribed to the (111) crystal plane of Ag (JCPDS No. 01-1167) [34]. This result indicated the introduction of metallic Ag on TiO_2_ to form Ag/TiO_2_, which was successfully synthesized via the photodeposition method.

The elemental compositions and chemical states of the Ag/TiO_2_ were subsequently characterized by XPS. The spectra of Figure 2a implied that the catalyst was mainly composed of three elements, including Ti, O, and Ag, while the C 1s peak might be attributed to the adsorbed CO_2_ or the contamination of hydrocarbons in the instrument [35]. In Ti 2p (Figure 2b) and O 1s (Figure 2c) spectra, peaks at 464.2 eV, 458.6 eV, 529.8 eV, and 531.8 eV were observed, which should be identified as the Ti 2p_1/2_ and Ti 2p_3/2_ spin orbitals of Ti^4+^ [36], lattice oxygen atoms, and oxygen atoms of surface hydroxyl [34], respectively. As for Ag 3d spectrum in Figure 2d, characteristic peaks of Ag 3d_5/2_ and Ag 3d_3/2_ appeared at 367.5 eV and 373.5 eV [37], which further confirmed the existence of metallic Ag in the catalysts.

SEM, TEM, and HRTEM were used to analyze the micromorphology of the as-prepared catalysts. As depicted in Figure 3a–c, Ag/TiO_2_ was made up of TiO_2_ irregular blocks with deposited Ag NP_S_. In accordance with the XRD results, typical lattice fringes with spacings of 0.170, 0.323, and 0.236 nm were observed, corresponding to the (105) plane of anatase, (110) plane of rutile, and (111) plane of Ag, respectively (Figure 3d). Moreover, the element mapping of Ag/TiO_2_ visually displayed its elemental distribution, where Ag was observed on the surface of the TiO_2_ block with uniformly distributed Ti and O. Based on the XRD, XPS, TEM, and mapping characterizations, it can be concluded that Ag/TiO_2_ was successfully synthesized by depositing Ag NPs over the TiO_2_.

### 2.2. Photocatalytic Inhibition Tests

#### 2.2.1. Inhibition Activities of Different Conditions

The photocatalytic inhibition activity of *A. flavus* over as-prepared Ag/TiO_2_ was investigated by a modified plate colony counting method. As displayed in Figure 4a, there were no inhibition activities in either darkness or light without catalysts; the inhibition rate of Ag/TiO_2_ under visible light irradiation was much higher than that in the dark, illustrating that the catalyst and visible light were necessary for photocatalytic inhibition activity. The inhibition rate of Ag/TiO_2_ was 27.58 ± 1.86% in the dark, which was attributed to the intrinsic disinfection effect of Ag NPs. It was also found that the composites presented a higher inhibition rate than the single catalyst under visible light irradiation in Figure 4a. Figure 4b indicated that the inhibition rate of *A. flavus* in the presence of Ag/TiO_2_ gradually increased with the increase in irradiation time. The inhibition rate of Ag/TiO_2_ against *A. flavus* was > 90% after irradiation for 15 min.

#### 2.2.2. Inhibition Activities of Ag/TiO_2_ with Different Silver Concentrations

In addition, the inhibition activities of Ag/TiO_2_ with different silver concentrations were explored. As demonstrated in Figure 4c, it was presented that the colony units in the presence of 1.5% Ag/TiO_2_ were the least compared with the others. The result was in accordance with the content in Figure 4d and suggested that 1.5% Ag/TiO_2_ has the highest inhibition rate of *A. flavus* under visible light. With the increase in silver concentrations, the photocatalytic inhibition rate of *A. flavus* firstly increased, followed by decreasing gradually when the silver concentrations were higher than 1.5% in Figure 4d. The reason for this phenomenon might be that too many Ag NPs generated a masking effect on the surface of TiO_2_ [38]. Therefore, the light-harvesting ability of the photocatalyst was reduced, which led to a reduction in the inhibition rate.

#### 2.2.3. Stability and Reusability of Catalysts

The stability of the catalyst was a decisive factor in long-term and continuous utilization, which was evaluated via the cyclic tests of 1.5% Ag/TiO_2_. As demonstrated in Figure 4e, 1.5% Ag/TiO_2_ maintained efficient photocatalytic activity during the consecutive runs, suggesting that Ag/TiO_2_ had good stability and reusability_._ The stability was further confirmed by the XRD pattern of the recycled 1.5% Ag/TiO_2_ (Figure 4f), which displayed no significant change after the fourth cycle. The above discussions verified that 1.5% Ag/TiO_2_ was potentially used in the photocatalytic inhibition of *A. flavus*.

### 2.3. Photocatalytic Inhibition in Peanuts

Peanuts were susceptible to *A. flavus* and aflatoxin contamination, which were regarded as some of the most serious problems for food safety [39]. Thus, the inhibition performance of *A. flavus* over the as-prepared 1.5% Ag/TiO_2_ on peanuts was evaluated. As shown in Figure 5a, the number of mycelia and spores on the surface of peanuts began to decrease gradually with the prolongation of irradiation time. Additionally, the contamination of *A. flavus* on peanuts was greatly inhibited after irradiation for 15 min. In addition, the concentrations of AFB_1_, AFB_2_, AFG_2_, and AFG_1_ were determined by the HPLC to analyze the contaminated level of aflatoxins in the above peanuts. As displayed in Figure 5b, the concentrations of aflatoxins on peanuts treated by photocatalysts under visible light were notably reduced. The corresponding inhibition rates of AFB_1_, AFB_2_, and AFG_2_ were 96.02 ± 0.19%, 92.50 ± 0.45%, and 89.81 ± 0.52%, respectively. These results illustrated that the 1.5% Ag/TiO_2_ can effectively inhibit *A. flavus* growth to control aflatoxin contamination in peanuts.

### 2.4. Peanut Quality Analysis after Photocatalytic Inhibition

The acid value (AV), peroxide value (POV), and the contents of fat, protein, polyphenols, and resveratrol after the treatment were estimated to investigate the effects of photocatalytic inhibition treatment on the quality of peanuts. AV and POV were two important indicators for testing whether peanuts were oxidatively deteriorated. The AV and POV before and after photocatalytic treatment were unchanged (Table 1), and they were lower than 3 mg/g and 0.4 g/100 g, which were the recommended values of the China Agricultural Industry Recommended Standard, respectively. The results of the correlational analysis suggested that photocatalytic treatment did not cause oxidation or deterioration of peanuts. Peanuts were rich in fat and protein [40], which belonged to six basic nutrients. The results showed the contents of fat and protein in peanuts were 51.64 ± 0.54% and 28.44 ± 0.76 before inhibition treatment, respectively. It has become apparent that the changes in fat and protein content were negligible after photocatalytic treatment. Polyphenols and resveratrol were nutrients with antioxidative functions in peanuts. Furthermore, it was found that the contents of polyphenols and resveratrol hardly changed after photocatalytic treatment. The above results suggest that the photocatalytic inhibition treatment could be a promising and efficient method to prevent *A. flavus* and to control aflatoxin contamination in peanuts.

## 3. Discussion

The above results had shown that the method was feasible and efficient in inhibiting *A. flavus* to reduce aflatoxin contamination in peanuts. The discussion of the inhibition mechanism helped to have an in-depth understanding and provided a theoretical basis and reference for the subsequent optimization of the method.

### 3.1. Photocatalytic Activity Enhancement Mechanism

The light-harvesting ability was one of the most significant factors for photocatalytic performance. Enhanced light-harvesting ability was conducive to photocatalytic activity. Thus, the UV-vis DRS absorption spectra of 1.5% Ag/TiO_2_ and TiO_2_ were recorded in Figure 6a. It is worth noting that the absorption range of Ag/TiO_2_ demonstrated an increase in the visible light region compared with pristine TiO_2_, which should be attributed to the SPR effect of metallic Ag NPs [41].

Electrochemical impedance spectroscopy (EIS) and transient photocurrent responses were employed to investigate the separation and transfer ability of the photogenerated carriers (h^+^ and e^−^). Generally, the higher the separation and the transfer efficiency of carriers, the better the photocatalytic activity. The smaller arc radius represented the smaller charge transfer resistance of as-prepared samples in EIS. Obviously, 1.5% Ag/TiO_2_ was the smallest one (Figure 6b), which indicated the most efficient charge-transfer property of the photocatalyst. Transient photocurrent responses were also measured to further observe the separation efficiency of carriers. As depicted in Figure 6c, the 1.5% Ag/TiO_2_ photocatalyst shows the strongest transient photocurrent response, implying the highest charge separation ability of 1.5% Ag/TiO_2_. 

### 3.2. Photocatalytic Inhibition Mechanism

To explore the active species in photocatalytic inhibition of *A. flavus*, radical trapping tests were performed. As shown in Figure 6d, benzoquinone caused a prominent decrease in inhibition rate, which indicated that •O_2_^−^ was the main active specie for inhibiting *A. flavus*. As for h^+^ and •OH, the addition of ammonium oxalate and *tert*-Butanol generated a lower loss of inhibition rate. Taking into account that •OH mainly originated from the oxidation of H_2_O driven by h^+^, the inhibition of *A. flavus* should be ascribed to •OH, which suggested that •OH was a secondary active specie. AgNO_3_ hardly had an effect on inhibition efficiency, indicating that e^−^ played the weakest role in this photoreaction system. The results of ESR tests were displayed in Figure 6e,f. Figure 6e showed that no •O_2_^−^ was generated from the 1.5% Ag/TiO_2_ in the dark, but it could be discovered that the •O_2_^−^ had a 1:1:1:1 quartet pattern characteristic after irradiation for 5 and 10 min. Similarly, as demonstrated in Figure 6f, the obvious characteristic peak of •OH was observed, which suggested the production of •OH radicals from 1.5% Ag/TiO_2_ under visible light irradiation.

To obtain a deeper understanding of the photocatalytic inhibition mechanism of *A. flavus*, morphological changes of *A. flavus* spores before and after the photocatalytic inhibition treatment were investigated through fluorescence-based live/dead fungal staining tests. In general, after staining by fluorescent dyes, SYTO 9 and PI, the living spores, which had intact cell membranes, would appear in green, while dead spores, which had damaged cell membranes, appeared in red [42]. As displayed in Figure 7a, almost all spore cells without visible light irradiation were alive. To the contrary, the number of dead spores increased considerably after irradiation for 15 min, indicating that photocatalytic treatment led to the damage of fungal cell structure.

Based on the above analyses, a possible photocatalytic inhibition mechanism of *A. flavus* over Ag/TiO_2_ was proposed and displayed in Figure 7b. When Ag/TiO_2_ was irradiated by visible light, the deposited Ag NPs on TiO_2_ improved the light-harvesting performance on account of the SPR effect, and they produced hot electrons. Hot electrons were transferred into the conduction band (CB) of TiO_2_ to create additional Fermi levels, which reduce the band gap energy near the CB [43]. Meanwhile, because of the intense interfacial contact between anatase, rutile, and Ag NPs, the conduction band potential of anatase was more positive than that of rutile. Additionally, the hot electrons firstly tended to flow to the CB of the rutile, followed by flowing to the CB of anatase [44]. This type of flow prolonged the lifetime of carriers, enhanced the separation and transfer abilities of electron-hole pairs, and was conducive to photocatalytic activity [45]. O_2_ was driven to reduction by e^−^ to •O_2_^−^, and h^+^ oxidized H_2_O/OH^−^ to •OH. These active species generated in the photocatalytic reaction attacked the spore cell membranes and caused damage to cell structures, which eventually led to reduced viability of *A. flavus* spores.

## 4. Conclusions

In conclusion, Ag/TiO_2_ composites were synthesized by a facile photodeposition method, which could efficiently inhibit *A. flavus* growth under visible light irradiation. Ag/TiO_2_ composites could also control *A. flavus* and then reduce the aflatoxin contamination in peanuts. Moreover, there were no obvious effects on the quality of peanuts after photocatalytic inhibition treatment. This efficient inhibition was due to the intrinsic disinfection and co-catalysis effects of Ag NP_S_ in composites. It was found that active species, such as •O_2_^−^, •OH^−^, h^+^, and e^−^, generated in photocatalytic reactions, could destroy the integrity of spore cell membranes to cause the reduced viability of *A. flavus* spores. This study presented an efficient, economical, and sustainable inhibition method for inhibiting *A. flavus*, which provided a reference for the green control of other pathogenic fungi.

## 5. Materials and Methods

### 5.1. Materials

All reagents, of analytical grade, were used directly without any further purification. AgNO_3_ was purchased from Guangdong Guanghua Sci-Tech Co., Ltd. (Guangdong, China). Tween 80, TiO_2_, and ammonium oxalate were acquired from Sinopharm Chemical Reagents Co., Ltd. (Shanghai, China). Benzoquinone and *tert*-Butanol were obtained from Sigma-Aldrich. Deionized water was used in all experiments.

### 5.2. Synthesis and Characterization of Ag/TiO_2_

#### 5.2.1. Synthesis of Ag/TiO_2_

Ag/TiO_2_ was synthesized by a facile photodeposition method [30]. In detail, 625 mg TiO_2_ powder was evenly dispersed in 200 mL methanol aqueous solution (10%) after sonication for 30 min, and then 868 μL AgNO_3_ solution (0.1 mol/L) was added drop by drop. After vigorous stirring for 1 h, the suspension liquid was irradiated with an ultraviolet halogen lamp (365–405 nm) for 1 h. The precipitate was centrifuged, washed three times with ethanol and water, and dried at 80 °C for 18 h. In addition, in order to evaluate the effect of silver concentrations on the photocatalytic inhibition performance, a series of Ag/TiO_2_ with different silver concentrations were prepared and labeled as 0.5%, 1%, 1.5%, 2%, and 2.5% Ag/TiO_2_ (Appendix A).

#### 5.2.2. Material Characterization

X-ray diffraction (XRD) analysis was recorded on a Bruker-AXS D8 X-ray diffractometer with a scan range of 20–80°. X-ray photoelectron spectroscopy (XPS, Thermo Escalab 250Xi, Waltham, MA, USA) was used to characterize the elemental compositions and chemical states of as-prepared Ag/TiO_2_. The morphology and microstructure were observed by scanning electron microscopy (SEM, Hitachi Limited SU8020, Tokyo, Japan), transmission electron microscopy (TEM, FEI Tecnai G2 F30, Hillsboro, TX, USA), and high-resolution transmission electron microscopy (HRTEM, FEI Tecnai G2 F30, Hillsboro, TX, USA).

### 5.3. Activation of Aspergillus flavus

All materials used in the process of the experiment were sterilized in advance [46]. The preserved spore suspension of *A. flavus* 3.4408 (China General Microbiological Culture Collection Center (Beijing, China)) was inoculated on AFPA Agar medium for activation, and it was cultured for three days (28 °C, 90% RH) until orange round colonies appeared. The activated mycelium of *A. flavus* was subsequently picked with sterile toothpicks and inoculated on Dichloran Glycerol (DG-18) Agar medium. After five to seven days of culture (28 °C, 90% RH), the spores were washed with Tween 80 (0.1%). Additionally, its concentration was estimated with blood counting plates under optical microscope. The obtained spores were stored in a refrigerator at 4 °C for later use.

### 5.4. Photocatalytic Inhibition Tests

#### 5.4.1. The Method of Photocatalytic Inhibition Tests

The modified plate colony counting method was used to evaluate the inhibition activities of Ag/TiO_2_. In detail, 25 mg of Ag/TiO_2_ powder was firstly suspended in 50 mL of spore suspension (10^6^ CFU/mL), followed by stirring for 30 min in the dark to reach sorption equilibrium. After that, the suspension was irradiated by a 300 W Xenon lamp (PLS SXE300, Beijing Perfectlight Inc., Beijing, China) with a visible light filter (λ > 420 nm). After given intervals (0, 3, 6, 9, 12, and 15 min), 1 mL of mixed liquid was taken as a sample, diluted with sterile water, and evenly coated on Malt Extract Agar medium. At last, the growing colony units were counted after culturing for 48 h at 28 °C. The inhibition rate (R) was calculated by comparing the colony number of *A. flavus* at the beginning (C_0_) and treatment time t (C_t_): R (%) = (C_0_ − C_t_)/C_0_ × 100%.

#### 5.4.2. The Effect of Ag/TiO_2_ with Different Silver Concentrations

According to the above method, photocatalytic inhibition tests were carried out with 0.5%, 1%, 1.5%, 2%, and 2.5% Ag/TiO_2_, respectively. Then, the inhibition performances were compared and analyzed.

#### 5.4.3. Cyclic Tests

The cyclic tests of photocatalytic inhibition of *A. flavus* were conducted to analyze the stability and reusability of the as-prepared Ag/TiO_2_ [47]. To be specific, the Ag/TiO_2_ composites were collected and reused by centrifuging, washing, and drying after each cycle.

### 5.5. Photocatalytic Inhibition of Aspergillus flavus on Peanuts

The peanuts with complete grains and uniform shapes were sterilized (121 °C, 30 min) and placed on Petri dishes (10 grains per plate). The spore suspension (1 × 10^4^ CFU/mL, 20 µL per grain) was inoculated on the surface of the peanuts and dried naturally. Subsequently, 200 µL of Ag/TiO_2_ dispersion (0.5 mg/mL) was evenly added to the surface of the contaminated peanuts, followed by irradiating with a Xenon lamp. After that, these peanuts were cultured for seven days at 28 °C. The growth situation of *A. flavus* on peanuts was observed. Whereafter, the treated peanuts were sterilized, dried at 105 °C, and ground for the determination of aflatoxins (AFB_1_, AFB_2,_ AFG_1_, and AFG_2_) using HPLC (Agilent 1100, Palo Alto, CA, USA). Specific parameters were shown in Table 2.

### 5.6. Evaluation of the Peanut Quality before and after Inhibition Treatment

In brief, the effects of inhibition treatment on peanut quality were investigated by determining acid value, peroxide value, and the contents of fat, protein, polyphenols, and resveratrol in peanuts before and after photocatalytic treatment. These indicators of peanuts were determined at the given times (1, 7, 14, and 21 days) after the treatment. The detailed detection methods of each indicator are in the Appendix A.

### 5.7. Study of the Photocatalytic Inhibition Mechanism of Aspergillus flavus

#### 5.7.1. Photocatalytic activity enhancement mechanism

The light absorption characteristics were analyzed through the UV-vis DRS (Shimadzu UV-3600, Kyoto, Japan). Photoelectrochemical properties were detected on a CHI 660E electrochemical workstation.

#### 5.7.2. Photocatalytic Inhibition Mechanism

Radical trapping tests were conducted to confirm the main active species in the process of photocatalytic inhibition, where benzoquinone (1 mM), *tert*-Butanol (1 mM), ammonium oxalate (1 mM), and AgNO_3_ (0.5 mM) were added to each parallel photocatalytic system to assume the role of scavengers for •O_2_^−^, •OH, h^+^, and e^−^, respectively [48].

ESR tests were employed to further explore the formation of reactive oxygen species. The production of •O_2_^−^ and •OH by the as-prepared Ag/TiO_2_ was quantified by ESR analysis (Bruker A300-10/12, Karlsruhe, Germany).

In addition, the fluorescence-based live/dead fungal viability assay was carried out using LIVE/DEAD^®^ FungaLight™ Yeast Viability Kit (Waltham, MA, USA) to explore the integrity of spore cell membranes and cell viability. The spore suspension was centrifuged, followed by washing before and after the inhibition treatment. Then, the resultant spores were stained for 30 min with SYTO 9 (6 µg/mL) and PI (6 µg/mL) in the confocal dish in the dark. Finally, the results were observed by a super-resolution confocal laser microscope (Zeiss LSM980, Oberkochen, Germany).

## Figures and Tables

**Figure 1 toxins-15-00216-f001:**
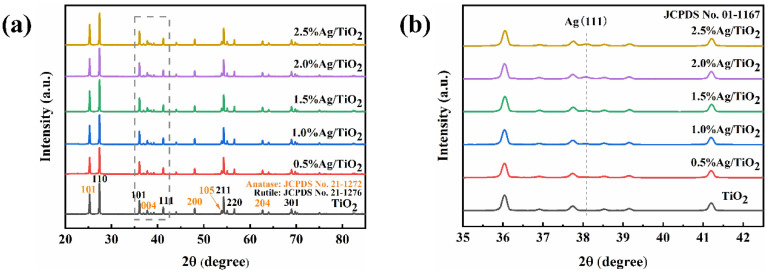
(**a**) XRD spectrum of TiO_2_ and Ag/TiO_2_ with different silver concentrations and (**b**) the enlarged pattern in 35–42.5° of Figure 1a.

**Figure 2 toxins-15-00216-f002:**
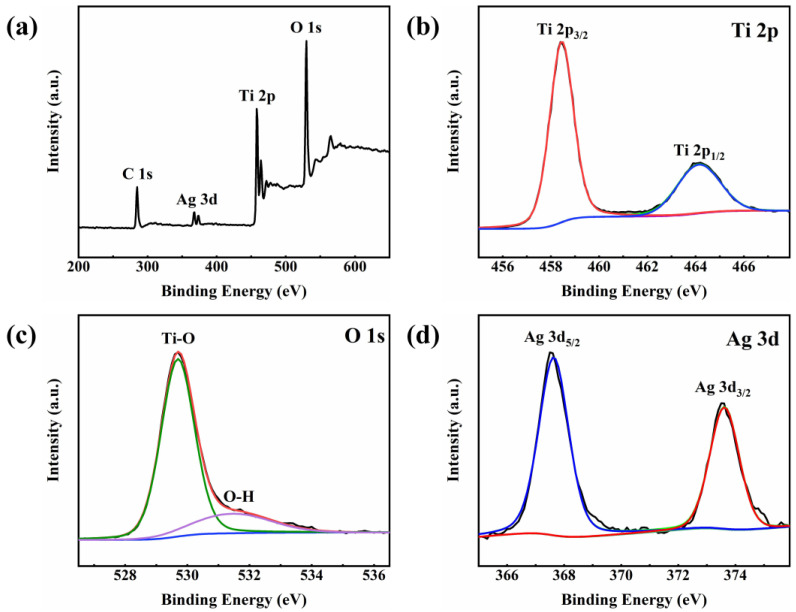
(**a**) XPS survey spectrum; high-resolution (**b**) Ti 2p spectrum; (**c**) O 1s spectrum; and (**d**) Ag 3d spectrum of 1.5% Ag/TiO_2_.

**Figure 3 toxins-15-00216-f003:**
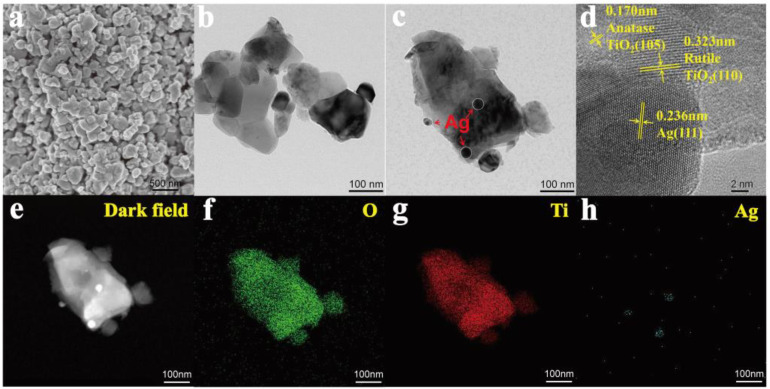
(**a**) SEM; (**b**,**c**) TEM; (**d**) HRTEM; the corresponding elemental mappings of (**e**) dark field; (**f**) O; (**g**) Ti; and (**h**) Ag of 1.5% Ag/TiO_2_.

**Figure 4 toxins-15-00216-f004:**
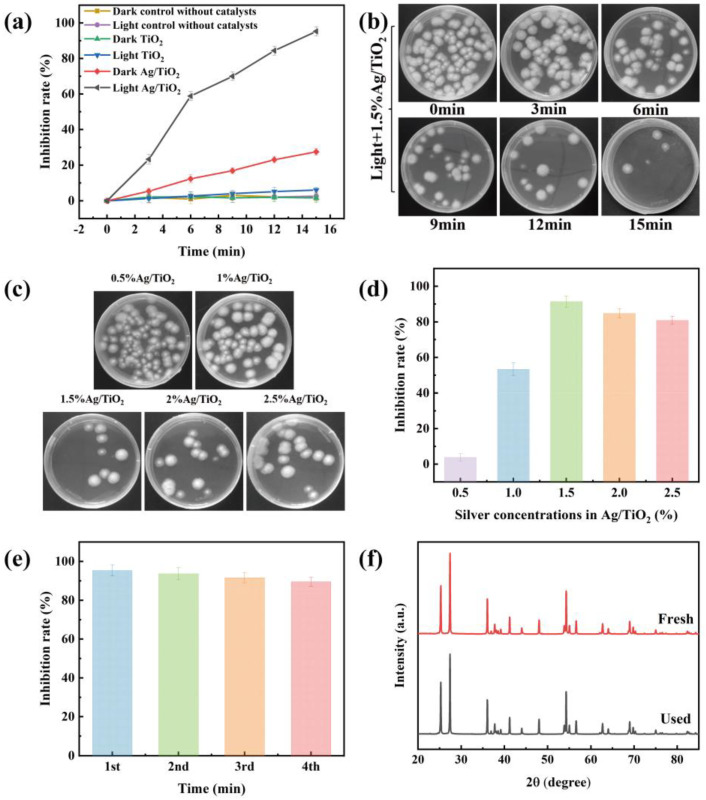
(**a**) The inhibition rates correspond to different treatment conditions; (**b**) the colony units under different irradiation times; (**c**) the colony units and (**d**) the corresponding inhibition rates over Ag/TiO_2_ with different silver concentrations; (**e**) the inhibition rates of four-cycle tests; and (**f**) XRD spectra of 1.5% Ag/TiO_2_ before and after four-cycle tests.

**Figure 5 toxins-15-00216-f005:**
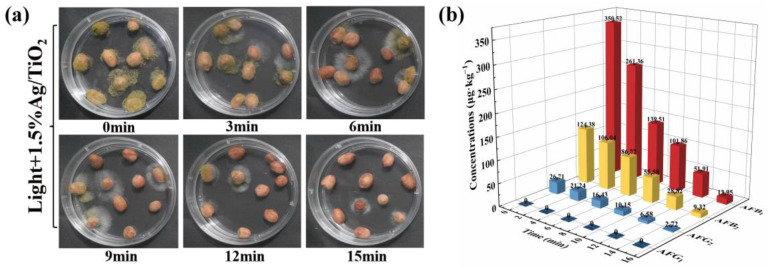
(**a**) The colonies of *A. flavus* on peanuts under different irradiation times and (**b**) the concentrations of aflatoxins in peanuts treated with 1.5% Ag/TiO_2_.

**Figure 6 toxins-15-00216-f006:**
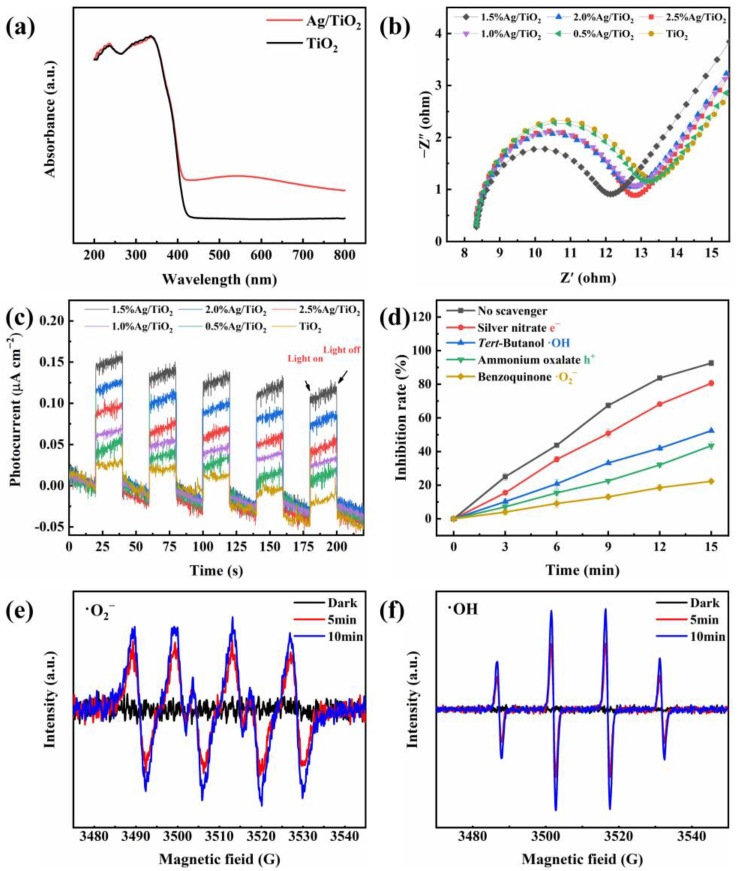
(**a**) UV-vis DRS spectra of 1.5% Ag/TiO_2_ and TiO_2_; (**b**) EIS and (**c**) transient photocurrent response spectra of TiO_2_ and Ag/TiO_2_; (**d**) the inhibition rates that correspond to different radical scavengers; ESR spectra of (**e**) •O_2_^−^ and (**f**) •OH for 1.5% Ag/TiO_2_.

**Figure 7 toxins-15-00216-f007:**
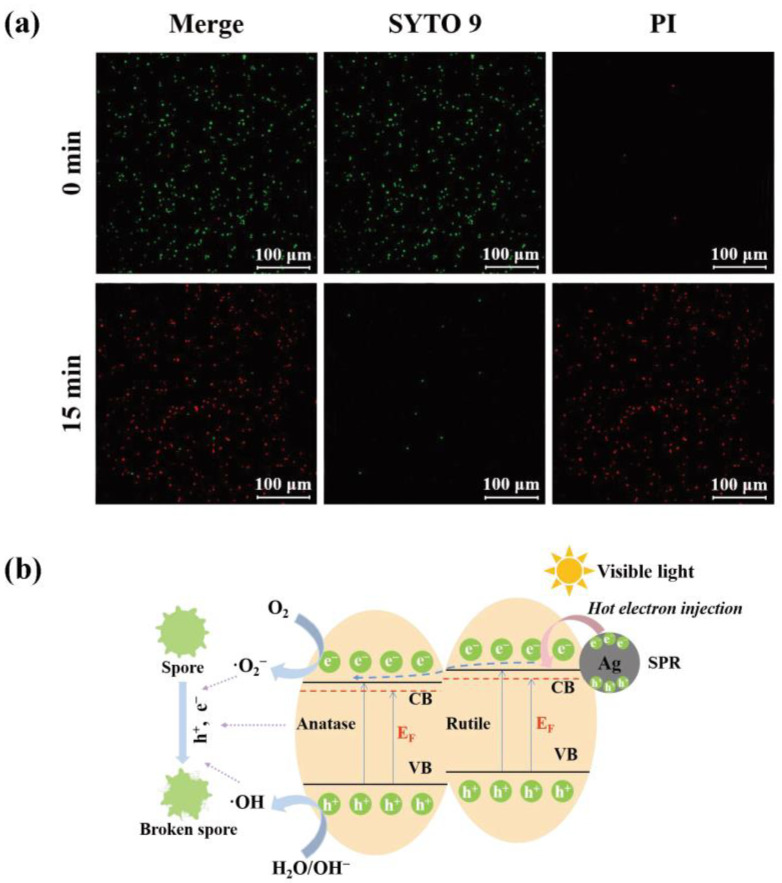
(**a**) Fluorescent images of live/dead spores of *A. flavus* and (**b**) the possible photocatalytic inhibition mechanism of Ag/TiO_2_.

**Table 1 toxins-15-00216-t001:** The determination results of peanut samples before and after photocatalytic treatment.

Peanuts	Acid Value (mg/g)	Peroxide Value (g/100 g)	Fat (%)	Protein (%)	Polyphenols (mg/kg)	Resveratrol (mg/kg)
Control *	1.46 ± 1.80	0.06 ± 1.41	51.64 ± 0.54	28.44 ± 0.76	27.48 ± 0.58	5.44 ± 1.15
1d	1.40 ± 2.88	0.06 ± 1.00	51.32 ± 0.35	28.32 ± 0.70	27.41 ± 0.67	5.35 ± 1.54
7d	1.45 ± 2.35	0.06 ± 1.38	51.49 ± 0.33	28.40 ± 0.29	27.45 ± 0.78	5.29 ± 1.01
14d	1.51 ± 1.74	0.06 ± 1.23	51.53 ± 0.43	28.42 ± 0.50	27.39 ± 0.56	5.24 ± 1.09
21d	1.47 ± 1.40	0.06 ± 1.18	51.48 ± 0.29	28.32 ± 0.46	27.33 ± 0.88	5.23 ± 1.06

* Control represented the peanuts without photocatalytic treatment. The form of data: mean of three values ± relative standard deviation.

**Table 2 toxins-15-00216-t002:** Specific parameters of the HPLC method.

Project	Parameters
Mobile phase	Methanol aqueous solution (45%)
Velocity of flow	0.80 mL/min
Chromatographic column	Sycronis C18, 4.6 mm × 150 mm, 5 µm
Column temperature	35 °C
Excitation wavelength	360 nm
Emission wavelength	440 nm
Injection volume	10 µL
Analysis time	14 min

## Data Availability

The data presented in this study are available in the article.

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
