# Peer review of "Efficient Inhibition of Aspergillus flavus to Reduce Aflatoxin Contamination on Peanuts over Ag-Loaded Titanium Dioxide"

_toxins, 2023, doi:10.3390/toxins15030216_

Round 1

Reviewer 1 Report

Overall, the article is interesting and well written. Information about the experimental set-up (UV lamp and origin of the mold strain) is missing. Several references should be corrected.

Reviewer 2 Report

The manuscript submitted to “Toxins” an MDPI journal entitled: “Efficient inhibition of Aspergillus flavus to reduce aflatoxin contamination on peanuts over Ag-loaded titanium dioxide” which discussed a green and powerful method of controlling Aspergillus flavus as well as aflatoxin contamination of peanuts over Ag-loaded titanium dioxide, the followimg points should be followed:

-          Abstract should be rewritten in more details and high lighting the main results in order to sound better and giving strength to the manuscript.

-          Introduction was written in organized manner but recent literatures are required.

-          Results and discussion should be separated into 2 chapters.

-          Figure 3: should be identified with more details.

-          Please add the ethical approval ID.

-          References:

·         Should be updated till 2023.

·         Self-citation ref. no. 21; should be exchanged to be fair enough to cite another ref. which will give more power to the results of your manuscript.

·         Add DOI to ref. whenever found.

Reviewer 3 Report

The paper is well-written, contains clear and relevant information regarding green, efficient, and economical ways to inhibit Aspergillus flavus and controlling aflatoxin contamination from peanuts. The introduction is brief and provides sufficient information on the importance of choosing the topic of this scientific paper.

The methods are generally appropriate, but I suggest including also, in the Material and method section, the working conditions for Aflatoxin determination (HPLC conditions and equipment). Also, it is mentioned that in supplementary files are presented the detailed detection methods for acid value, peroxide value, and the contents of fat, protein, polyphenols, resveratrol in peanuts before and after photocatalytic treatment. I suggest including this methods in the manuscript.

I suggest that under Table 1. The determination results of peanut samples before and after photocatalytic treatment, to explain in what form the data in the table are represented (mean of 3 values + standard deviation)?

This work presents interesting results that are clear and compelling; the authors making an important contribution to the research literature in this area of investigation.
